# The Wooden Architecture Route as an Example of a Regional Tourism Product in Poland

**Małgorzata Zdon-Korzeniowska \*** and **Monika Noviello \***

Department of Entrepreneurship and Spatial Management, Institute of Geography, Pedagogical University of Cracow, 30-084 Cracow, Poland
**\*** Correspondence: malgorzata.zdon-korzeniowska@up.krakow.pl (M.Z.-K.); mnoviello@up.krakow.pl (M.N.)

**Abstract:** In the modern economy, there is a significant increase in interest in tourism, both at the level of states, regions, communes and individual places. Tourism is seen primarily as an opportunity for economic development, but also for social development and activation of local communities. Well-managed tourism can become a way to preserve and protect the natural, cultural and historical heritage of specific places or regions by exploring and nurturing it. Heritage elements become, on the one hand, attractions around which unique tourism products are created, and on the other hand, a kind of distinguishing feature of a given place or region, based on which local communities build their identity and sense of belonging. The concept of creating regional tourism products could integrate these two factors, i.e., tourism and heritage. The article presents the concept of a regional tourist product on the example of the Wooden Architecture Route (case study).

**Keywords:** heritage; heritage interpretation; Poland; regional development; sustainable tourism; tourism

## 1. Introduction

Many countries identify in tourism an opportunity of socio-economic development. It is perceived as an effective tool within the policy of countries and regions to equalize the differences in their socio-economic development. Tourism is often regarded as an important factor not only for the development of touristically attractive areas or regions, but also for the activation of territories where the function of tourism has not yet played a significant role, if any at all. Tourism-related activities become a supplement or an alternative to unprofitable agricultural activities, unsuccessful health-resorts or industrial agglomerations. As an ecological and environmentally friendly "industry", tourism is an alternative to other developmental trajectories, the choice and application of which in a given area would result, for various reasons (for example ecological), disadvantageous or simply not possible.

In addition to stimulating the socio-economic development of towns and regions, tourism also contributes to the exploration and cultivation of the cultural and natural qualities and resources of such sites. Well-managed tourism can even become a way of preserving and protecting the natural, cultural and historical heritage of given areas or regions through its discovery and cultivation. Heritage elements are, on the one hand, attractions around which unique touristic products are formed, while on the other hand they become a sort of identity traits of a given place or region, based on which local communities are building their own identity and sense of belonging.

The concept of creating regional tourism products consists of a methodological approach integrating these two factors, i.e., tourism and heritage. In recent years, different kinds of touristic routes turned into a popular form of presenting and interpreting the local and regional heritage. They become an essential factor in initiating social and civic change at the local level [1]. The Wooden Architecture Route (WA Route), which is the subject of analysis of the present study, was chosen as an

example of regional tourism product since as demonstrated by the research, touristic routes based on a given local heritage are most frequently indicated as examples of regional tourism products [2]. The second important reason for choosing the WA Route as an example of regional tourism product is the fact that about 10% of historical monuments in Poland are constituted by wooden buildings and transom structures. Retaining such objects lies in the interest of society for the sake of their historical, artistic and scientific value [1]. The issues associated with the creation and the development of tourism products with the use of the historical, cultural or natural heritage of a given place are important from the point of view of the local and regional development because these products themselves can be thought of as a kind of activation tools of development.

## 2. Materials and Methods

The purpose of the article is to present the concept and the way of development of the regional tourism product in the case of The Wooden Architecture Route operating in four provinces in Poland. The aim of the present work is to illustrate, based on source literature, the conception of regional tourism product as well as its definition. However, for the realization of the latter task, the following guidelines were assumed:

- the general, territorial (spatial) way of presenting and defining the tourism product is the closest to what the tourist understands as "product" therefore the entire experience associated with the arrival and stay in a given place;
- within the framework of the integrated regional tourism products occurs the activation of the development of the region and the local communities through tourism in accordance with the principles of sustainable development and the protection of the heritage of the region through its cultivation and preservation for future generations;
- in the process of creating and shaping a regional tourism product there are several subjects involved: the private, public, and non-governmental sectors as well as the local community.

On the other hand, in part, the empirical work consists of an analysis of the idea of the creation of the WA Route, the structure of the itinerary and its functioning. An analysis of the marketing activities undertaken in connection with the shaping of the tourism product of the WA Route was as well carried out. The primary research method applied is the case study. It's a qualitative research method, whose main purpose is the mostly thorough illustration of a certain phenomenon, object, action, or in accordance with the name-specific case. According to A. Lee [3] it is comparable to the natural experiment. Such method could be applied in many research fields it is often used, for instance, in law or medicine. Since the 90ies of the last century it is applied in Management sciences as well [4–6] both as a scientific method and a method of Management education [7] p. 8.

W. Gregorczyk, with reference to Management sciences, defines the case study as an in-depth detailed description, usually real, of a given economic phenomena e.g., the organization, the management process, its elements or business environment, in order to provide conclusions about the causes and results of its course [7] p. 10. As the author indicates, this method is empirical because it analyses and evaluates real-world phenomena and is used especially for descriptive research topics. It gives the answer to the question what, where and how did a given phenomenon happen. This type of scientific research focuses more on an in-depth understanding of the phenomenon than on the analysis of variables. R. Yin recommends using the case study method to find answers to questions that are revealing, and therefore concerning the "how" and "why" a given phenomenon occurs [8] p. 8.

For the needs of carrying out the case study, various techniques and tools for collecting and analysing data are used. These can be observations, participatory observations, interviews, surveys, documentation of the organization under study, press sources, Internet sources, databases run by various institutions, etc. [7] p. 10. Therefore, for the purposes of this study, in order to provide a rich information material for a more detailed analysis other complementary testing methods have been used, namely:

- desk research-study of existing materials: folders, multimedia presentations, maps, guides and other promotional materials issued by Marshal's offices of the provinces through which the WA Route runs, as well as the internal materials of such offices on the WA Route;
- participant's observation it consists on the planned gathering of observations on the functioning of the WA Route from the point of view of the consumer-tourist visiting the WA Route.

## 3. Literature Review

### 3.1. Regional Tourism Product—Nature and Definition

The idea of using the heritage and culture of given places or regions for their sustainable development has been present in source literature for several decades [2,9–20]. Regional tourism product is a concept used both in the scientific literature on tourism [21–23], as well as in the economic practice (Regional Tourism Product Development Program, Guidelines 2017–2018 Financial Years). Such notion is understood quite diversely, as confirmed by Zdon-Korzeniowska (2009). The regional tourism product is regarded as either a specific area (town, region), a tourist route or a local event, as well a single product constituting a product of the local food processing, a craft tradition or a material element of the culture belonging to a given place.

Regional (lat. regionalis) means as follows: "Associated with a given region, characteristic of, known, used, occurring in a specified region, originating from a given region" [24]. The essence of the regional tourism product is therefore related to its close connection with the given region, not only by the fact that it is located in the area of such region. This relationship is primarily marked by reference to the idea of regionalism, to the spiritual and material heritage of a given region the historical, cultural and natural heritage. The regional tourism product is characterized by its peculiarity, dissimilarity and uniqueness. It is an original product, strongly identified in the region in which it is offered, typical of the latter, referring to its identity and idea of regionalism. The characteristics of the regional tourism product are attributable to the origin of a given region. It expresses close ties with the region in which it is offered, with the existing cultural and natural conditions and resources. The idea of the regional tourism product is to meet the needs and expectations of tourists visiting a given place, benefitting from the resources of the region as well as from the historical, cultural and natural heritage with the simultaneous preservation of such heritage without any damage for current and future residents.

The essence of the regional tourism product is therefore its connection to the region, which is mainly evident through the "regionality" of the tourist attractions, on the basis of which the product is created. Regional attractions are related to a given place (region) in a "permanent" manner, not temporary, non-occasional, "not migratory", based on the resources of such place (region), authentic, exceptional, in a specific form occurring only and exclusively in that given place (region). Regional attractions are those based on the unique resources of the region, which are characterized by naturality and uniqueness. In relation to the mentioned traits they determine the activity of the tourists (historical, intellectual, scientific activities), stimulating them regardless of the distance to visit a given place. These attractions are non-reproducible, constituting a natural and cultural element, as well as historical, of the heritage of a given region.

### 3.2. Heritage as the Basis for Creating a Regional Tourism Product

Heritage is "something" passed down from generation to generation, our legacy; what we are currently living and what we hand down to children and grandchildren. These are our identifiers, reference points, our identity [25] p. 12. R. Hewison (1989) defines heritage as "what past generations preserved and handed down to us, and what a significant amount of the population is willing to secure for future generations" [26]. The core of the regional heritage is therefore the cultural and natural values of the region passed down from generation to generation [27]. Within the meaning granted by the International Convention for the Protection of the World Cultural and Natural Heritage adopted in Paris in 1972, "cultural heritage" is considered to be:

- monuments: works of architecture, works of monumental sculpture and painting, elements and structures of archaeological importance, inscriptions, caves and the concentration of such elements, having a unique universal value from the historic, artistic and scientific point of view;
- groups of separate or associated structures which, due to their architecture, uniformity or integration with the landscape, present a unique universal value in terms of history, art or science, and
- historical places: works of man or conjoint works of man and nature, as well as zones, archaeological sites, which present a unique universal value from a historical, aesthetic, ethnological or anthropological point of view.

Whereas "natural heritage" is considered to include as follows:

- natural monuments originated by physical or biological formations or the concentration of such formations, showing a unique universal value from a scientific or aesthetic standpoint;
- geological and physiographic formations as well as zones of precisely definite borders, which constitute habitats for endangered species of animals and plants and which possess an exceptional universal value from the scientific or behavioral point of view;
- places or natural zones of strictly defined borders, possessing a unique universal value from the point of view of science, behavior or natural beauty.

As indicated by G.J. Ashworth (1994), the use of historical, natural or cultural heritage of given places or regions to express and emphasize their identity is adopted to build and strengthen the spatial policies of these areas. Therefore, according to the concept of regional tourism product, the heritage (historical, cultural, natural) of a given place (region) incorporates primarily two functions: the regional attractions (original, unique) around which the tourism products are built, and the characteristic identity of a place (region) based on which the region's image is created, the local communities mobilize and build their sense of regional identity. In doing so they foster the same efficiency and effectiveness conducted by local authorities and regional development policies [28]. Furthermore, by nurturing, cultivating and discovering the heritage, the latter is preserved for future generations.

"Heritage and tourism are inextricably linked. The heritage attracts tourists, and in turn tourism directs attention to the heritage" [29] p. 139. Nevertheless, as G.J. Ashworth points out, "the relationship between them is not harmonious, and achieving a state of lasting harmony requires active management" [30] p. 167. Moreover, the heritage serves to meet many needs, and tourism is just one of them. Using the elements of the historical, cultural or natural heritage for the creation of tourist attractions and further the tourism product, it is important to keep in mind as indicated by D. MacCannel (2002) to maintain their authenticity, so that they are not just distorted interpretations, which aim to solely excite the greatest interest among tourists [31]. According to M. Karczewska (2002), the construction of tourism products based on heritage elements generally finds its meaning only in a defined terrain, and the only subject capable of "understanding" such heritage and therefore introducing it into the tourist offer is the local community, which is settled in the heart of this heritage. As the author emphasizes, only the local population can mobilize and cause the heritage to revive and become a part of the local development. It is therefore essential that the inhabitants of a given region are able to identify it and recognize its unique value [32]. The local community is one of the key actors in creating regional tourism products. On the other hand, according to J.R. Ritche and M. Sinsa's (1978) classification of tourist attractions, the local population itself can also be considered as a tourist attraction, which by cultivating a "living" tradition of folklore or dialect becomes a carrier of immaterial resources and of the heritage of the region. The hospitality and openness of the local community, as well as its positive attitude towards visitors are also contributing factors helping to raise the attractiveness of the regional tourism product. It is often the inhabitants their hospitality, culture and tradition, which become the main tourist attraction attracting tourists to a given place. It is considered that "the tradition and cultural diversity of the various social groups or geographical regions are factors which strongly influence the behavior of purchasers" [33].

Assuming that the regional tourism product is a category of spatial product and therefore a product offered by a certain area, in its definition we cannot limit ourselves to an indication of the attractions, although they constitute the point of gravity of each tourism product. However, it is worth mentioning the relevance of all the infrastructures and services, which enable tourists to benefit from the allure of the site's attractions, such as appropriate tourism development, security level, commercial and service network or other facilities with market or non-market characters. Admittedly, these para-tourist services and products are merely a means of achieving the basic purpose of a tourist's arrival to a given place, yet their level and structure may be decisive when it comes to pick the destination of the journey (place of destination) [34].

Although, as already mentioned above, the nature of the tourist product presupposes the regional character of its attractions, this is undoubtedly not the only element that should indicate the regionality of the product. The "regional" nature of the product ought to be reflected also in other components. Due to the regional features visible in the interior design, in architecture, music, kitchen-restaurants, hotels or other elements of the infrastructures, they become more than places where to sleep and eat. They are a source of aesthetic and emotional experiences. To recapitulate, we can assume that the regional tourism product is a category of spatial product, thus a tourist product offered by a delimited area, built on the basis of regional attractions its natural, cultural and historical heritage in a particular form that occurs only in a given location.

## 4. Result and Discussion

### 4.1. AD Route as a Tourism Product—The Genesis and Idea of the Route

The Wooden Architecture Route, subject of the analysis of the present work, currently crossing the areas of four provinces of southern Poland (i.e., the voivodeships of Lesser Poland, Subcarpathian, Silesian and Świętokrzyskie), corresponds (at least in the assumptions adopted in this study) to the wide perspective defining the Regional Tourism Product Development Program product as a spatial one [35]. Therefore, it integrates the role of many subjects involved in its activities, built based on regional attractions (buildings of wooden architecture), testifying not only the region's history of construction, but also the fates of the inhabitants of these lands (Figure 1). These attractions are an essential part of the historical and cultural heritage of the region.

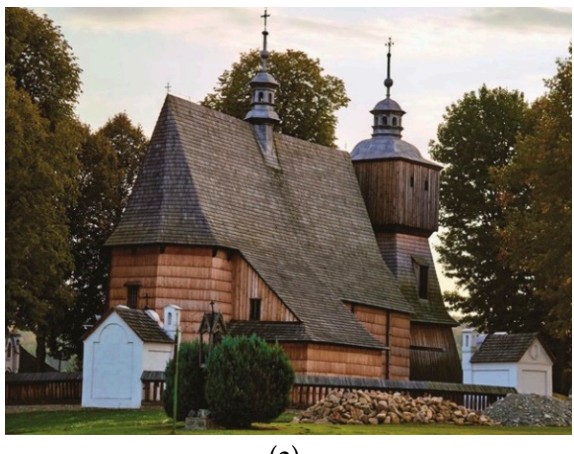　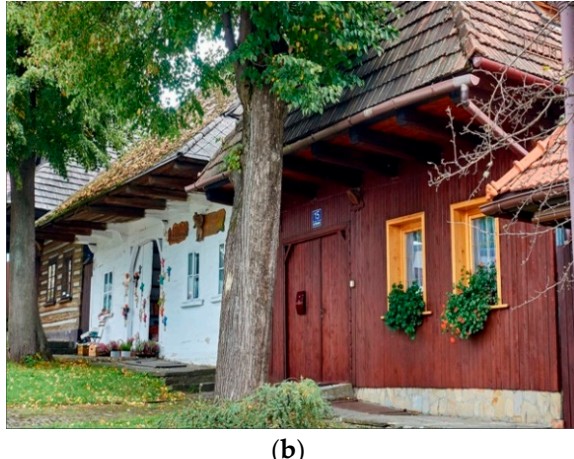

(**a**)　　　　　　　　　　　　　　　　　　　　　　　　(**b**)

**Figure 1.** Wooden Architecture Route: (**a**) The Gothic church of All Saints in Blizne (UNESCO World Heritage List); (**b**) The Lanckorona Urban Layout and Buildings (Source: Author's own archive- Photo M. Szczepek (**a**) and A. Korzeniowski (**b**)).

The first basic purpose of the WA Route was to protect the sacral architecture of wooden buildings, and only later it evolved towards the idea of transforming the route into a tourism product. In this

aspect, an important role in the genesis of this endeavor was played by the figure and activity of a Cracowian scholar, Dr. Marian Kornecki, an art historian, but especially an eminent connoisseur and indefatigable stock taker of monuments of wooden architecture. She was author and co-author of many publications and scientific studies on wooden architecture, and the initiator of the idea of protecting and preserving wooden sights. Currently, M. Kornecki is the patron of the Małopolska voivodeship's award, granted for outstanding achievements in the field of preservation and promotion of monuments of wooden architecture.

The main purpose for the creation of the WA Route was, therefore, to protect and preserve the cultural and historical heritage of Lesser Poland the popularization of a unique heritage of wood architecture on a national and European scale, but also on a world one, and only as an additional goal to build a tourism product on the WA Route. The delimitation and marking of the trail was intended to encourage the local community and the tourism industry to further develop it and create a complete tourist offer. The idea itself was "born" at the end of the 90-ies of the past century in the Cracow Tourism Development Agency (pl. KART). It concerned initially the integration of about 40 wooden sights into the tourist route.

From its inception, in the assumptions of originators, the Wooden Architecture Route was supposed to have the form of a tourism product, cultivating the legacy of the wooden constructions. In 1999 the idea of a Wooden Architecture Route was presented by one of the originators and first animators of the Route, Edward Turkiewicz at the time president of the Cracow Tourism Development, as follows: "The route will provide the opportunity to discover ( . . . ) the unique mosaic of wooden architecture varieties, to reveal its beauty and rarity. ( . . . ) the proposed route will not only be a sightseeing suggestion. Around the trail should ensue hotels and guesthouses, tourist shelters, bars and restaurants, maybe could return handicrafts in their natural form and surroundings. A similar situation arises around all tourist attractions and areas for the development of various forms of active tourism: equestrian, skiing, bicycles, etc. ( . . . ) The propositions will be addressed to the different subjects interested in participating in the project. The project will succeed if everyone recognizes that the development of tourism in this area should be supported, and that tourism is a source of income, by co-financing the various stages" [36] p. 238.

After the interruption of the Cracow Tourism Development Agency's functions (KART) and after the administrative reform in 1999, when the provincial government was separated, KART's idea, i.e., the creation of touristic routes based on the sights of wooden architecture was maintained by the Marshal of the Małopolska region. This idea the integration in the tourist route of initially 40 wooden sights was consulted with local authorities, which proposed also other projects regarding the wooden architecture sights, which could be included in the Route. The Małopolska initiative to create the WA Route has been expanded to two neighboring voivodeships as well: Subcarpathian and Silesian. On the 17 April 2002, in Sanok, a collaboration agreement for the realization of the Wooden Architecture Route was signed between the marshals of the Lesser Poland, Subcarpathian and Silesia voivodeships, according to which the route is intended for the promotion of the voivodeships, the development of cultural tourism and the protection of the national heritage [37]. The parties have undertaken in particular to:

(1) the mutual transmission of information and experience resulting from work on the implementation of the Wooden Architecture Route;
(2) an agreement to present the financial resources;
(3) the application of a common marking of the Wooden Architecture Route;
(4) the creation of marketing and promotional policies related to the realization of the tourism product.

Moreover, the commitment of the voivodeships included, each on one's own, the realization of a creative documentation of the Wooden Architecture Route based on the guidelines adopted in the Lesser Poland's voivodeship. Each of the voivodeships, also on their own, has taken action in relation to the delineation and the marking of the WA Route, as well as its further maintenance. In order to

accomplish such tasks, it was necessary to make arrangements with territorial government units at a local level in each region and with the stewards and holders of historic buildings.

Therefore, the Subcarpathian and Silesian Voivodeships adopted from the Lesser Poland voivodeship the general idea and way of realization for the creation of a tourism product in the form of the Wooden Architecture Route. Pursuant to the agreement, the Lesser Poland voivodeship received the role of coordinator of the whole project. Each of the provinces had also to establish regional coordinators for the effective implementation of the objectives and tasks enshrined in the agreement. They also identified the areas of cooperation and the way they ought to be organized between the marshal offices. Another province that has lengthened the WA Route was the Świętokrzyskie Voivodship. All the ideological assumptions of the Wooden Architecture Route are in line with the concept of regional tourism product. The WA Route was supposed to have ultimately the form of a full, subordinated to a common concept, tourism product, cultivating the historical heritage of the region and integrating the activities of many subjects.

### 4.2. AD Route—Course and Structure

The WA Route is an enterprise with a superregional character, which, beyond the Lesser Poland voivodship from which started the whole initiative of the Route, was joined, as mentioned above, by three more voivodeships: Subcarpathian, Silesian and Świętokrzyskie. The WA Route is also connected to a similar project in the country of Prešov (Prešovský Kraj) in Slovakia. The WA Trail was designed as a car route, mainly for motorized tourists. He currently measures about 4262 km long and connects on its route 532 historic wooden sights of exceptional cultural value, located in different municipalities (Table 1). In each region (voivodeships), the WA Route is divided into smaller segments: "itineraries" or "loops", in total it is split in 24 itineraries.

**Table 1.** Wooden Architecture Route-general quantitative information.

| Detailed List Voivodeship | Number of Sights | Total Lenght of the Itinerary (km) | Number of Itinerary's Segments |
|---|---|---|---|
| Lesser Poland | 253 | 1500 | 4 |
| Subcarpathian | 127 | 1202 | 9 |
| Silesian | 93 | 1060 | 6 |
| Świętokrzyskie | 59 | over 500 | 5 [1] |

[1] own elaboration based on [38–41].

The first step in the process of creating the WA Route was the design. It consisted in the development of the concept and in the outlining of route paths based on the identified wooden architecture sights and existing road infrastructures within the region. The rationale behind the WA Route was to emphasize the existing natural and cultural values of the voivodeship, and therefore create potential opportunities to complement the isolated itineraries, which are specific of a given subregion, with additional attractions. In the development of the concept of the WA Route it is assumed that it will become the axis of a full product offer, which means that has to be complemented with lodging and catering, as well as other additional attractions. The course of the WA Route in individual voivodeships is presented in Figure 2.

After the elaboration of the concept of the WA Route, as well as the agreement on its course with local authorities, the Lesser Poland voivodeship, in cooperation with the Academy of Fine Arts in Cracow, announced a competition for the realization of the graphic symbol of the Wooden Architecture Route. Selected within demanding competitions, the project has become the official logo of the WA Route, which is still in operation for the terrace of the Route running in the other remaining voivodeships.

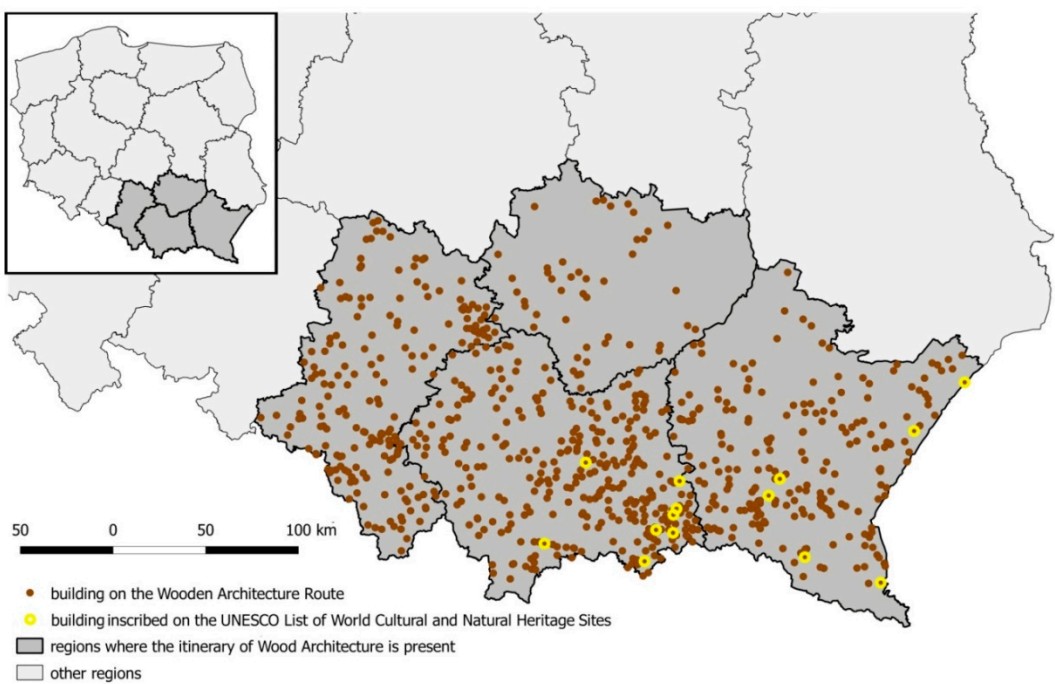

**Figure 2.** Territorial scope and location of individual buildings of the Wooden Architecture Route in the voivodeships of Lesser Poland, Subcarpathian, Silesian and Świętokrzyskie (Source: Author's own elaboration).

The Wooden Architecture Route is one of the biggest enterprises related to the marking out of the touristic routes in Poland. Not only because of its territorial reach, but also for the subjective scale and the planned time horizon. It is also the only initiative in Europe that much extensive in terms of diversity and multiculturality included in various parts of the route. It covers a wide spectrum of wooden artistic and architectural monuments. Among the sights included in the WA Route are present: churches, orthodox churches, open-air ethnographic museums, manor houses, whole urban systems (such as rural and small-town buildings), chapels, roadside shrines, bell towers, homes, rural cottages and noble manors, wooden farm buildings (like granaries or water mills), taverns, forester's lodges, palaces and open-air ethnographic museums. Some of them are real "pearls" of architectural art 14 of which were included in the UNESCO World Cultural and Natural Heritage list. These are located in the Lesser Poland voivodeship and Subcarpathian voivodeship (Table 2).

**Table 2.** Wooden Architecture Route on the UNESCO list-general quantitative information.

| Name | Location | Time of Construction |
|---|---|---|
| **Lesser Poland Voivodeship** | | |
| Saints Philip and James Church | Sękowa | approx. y. 1520 |
| St. Michael Archangel's Church | Binarowa | approx. y. 1500 |
| St. Michael Archangel's Church | Dębno | the second half of the 15th century |
| St. Leonard's Church | Lipnica Murowana | the end of the 15th century |
| St. Michael Archangel's Church | Brunary | end of the 18th century |
| St. Paraskevi Church | Kwiatoń | the second half of the XVII century |
| Protection of Our Most Holy Lady Church | Owczary | about middle XVII century |
| St. James Church | Powroźnik | the beginning of the XVII century |
| **Subcarpathian Voivodeship** | | |
| Assumption of Holy Mary Church | Haczów | from the first half of the XV century |
| All Saints Church | Blizne | middle XV century |
| St. Michael Archangel's Church | Smolnik | the end of XVIII century |
| St. Michael Archangel's Church | Turzańsk | the beginning of the XIX century |
| Mother of God Church | Chotyniec | the beginning of the XVII century |
| St. Paraskevi Church | Radruż | the end of the XVI century [2] |

[2] own elaboration based on [38,39,41].

The reasons for the entry of the churches in the UNESCO World Heritage List are to be found primarily in the authenticity of the sights: the authenticity of architecture, material and construction, but also the authenticity of the functions, contents and rituals which are maintained by the conservators.

Initiatives consisting in building tourist routes as tourist attractions of wooden constructions could also be found in other regions of Poland. An example may be the "Wooden route of sacral constructions" in Opole Silesia, leading along special marks from Opole to Olesno. It is a common route for cycling and car-coach tourism, which allows to visit twelve old wooden churches. The initiative related to the conservation and preservation of wooden architecture objects was also undertaken in Pomerania. The Sightseeing Commission of the Szczecin Regional Branch PTTK them. Stefan Kaczmarek, hoping to "save from oblivion" sights of wooden architecture, appointed the sightseeing badge "Wooden Architecture Route in Poland" [42]. The badge has three degrees (brown, silver and gold), and the condition for its acquisition is to visit the appropriate number of sights of wooden architecture (Wooden Architecture Route in Poland. Sightseeing badge terms and conditions). Such initiatives are now also carried out in relation to the WA Route in the voivodeships of Lesser Poland, Silesia and Świętokrzyskie.

## 5. Conclusions and Recommendations

The issues concerning the nature, construction and shaping of the tourism products is now one of the most frequently discussed topics of scientific exploration in the field of tourism, as well as the interest of general governments, authorities and other subjects responsible for shaping tourism.

The importance of such issues seems to result from a few conditions:

- Tourism is now one of the most rapidly growing economic areas and as an economic phenomenon constitutes an important factor for the activation of developmental policies of countries, regions, municipalities, towns and villages not just those attractive from a touristic viewpoint.

- Even the European Union, aware the multifaceted nature of tourism, recognizes the latter as the driving force behind the development of countries at all levels. Tourism is seen as not only an important factor for economic growth, but also as an area which could contribute to the protection of the environment and the preservation of the cultural heritage, the promotion of democracy and political balance (Tourism and development, resolution...), ensuring coherence and balance between the regions. The European Union emphasizes the preservation of the principles of sustainable tourism, stressing the need to support the protection and preservation of the environment, heritage and cultural identity, as well as the conservation and promotion of local traditions, specialties, and support for local communities. The best practices and environmental performance indicators in the tourism industry have been developed by experts from the European Commission and published in the form of the Commission Decision (EU) 2016/611 of 15 April 2016 concerning the document referring to the tourism sector [43].
- The United Nations has announced the year 2017 as the International Year of sustainable Tourism for Development. The UN initiative aims to promote changes in state policies, business strategies and consumer behavior in the construction of a sustainable tourism sector.
- The negative effects of the development of mass tourism and the awareness of their consequences for the further development of the tourism business and the tendency to counterbalance such development on a global scale contributed to search for alternative, more ecological forms of tourist activity. The essence of the constituting alternative tourism includes not only respecting and exploring the local cultural heritage, but promotes also the reconstruction of the latter in both the material and spiritual sphere [44]. The creation of tourism products based on the heritage of the region contributes to their discovery and cultivation, thereby preserving it for future generations according to the concept of sustainable development. It also gives a chance to stand out by creating a unique offer.
- The institutional economic transformation, as well as the reform of public administration, has led in Poland to a decentralization of the economic governance, thereby increasing the importance and the activity of self-government structures in shaping the socio-economic processes and phenomena. It followed the articulations of policies at a local and regional level [45] in which tourism started increasingly to be perceived as an opportunity for an economic revival of the regions, encouraging entrepreneurship and stimulating local communities. The mentioned changes have also initiated the process of rebirth of a sense of cultural identity in individual regions, which is one of the essential factors affecting the social activity of a given place (region) [46].

Regional tourism products, among which we can acknowledge the WA Route discussed above, constitute an excellent tool for local and regional development and the activation of local communities. However, they require constant monitoring and improvement based on marketing research. To solely create an offer based on, for instance, the appropriate adaptation of the sights, the marking or routing of the trails is important, but not sufficient. The condition for an effective commercialization of such products is i.e. making it available to tourists, which should be in our times examined and implemented in a multi-faceted manner through improvements connected with the possibility of a physical journey, including convenient opening hours, language and information availability and their accessibility in virtual spaces. The analysis of the activities and the marketing tools used in terms of the shaping tourism product WA Route indicates the advantage of activities which are promotional and in favour of shaping the image of the product compared to those associated with the creation of a full, integrated offer of the tourism product WA Route or related to its availability. Moreover, in order to respond to the needs and expectations of the tourists, it is necessary to examine their views as well as the opinions and expectations of the local community, whose degree of satisfaction and commitment to create a tourist offer of the place often influences directly the quality of such offer and hence the satisfaction of the tourists. In relation to the management of the WA route, such studies have not been conducted.

An important element testifying the good management of a touristic spatial product in which we acknowledge the WA Route is the cooperation and coordination of activities between all the

contributors to the product. In this respect, action is needed to activate and sustain the cooperation of the entities that make up the tourism product discussed. Cooperation with the subjects at the disposal of which are to be found the wooden architecture objects, however difficult not only because of the multiplicity of such entities, but also as a result of their insufficient or often low involvement seems to be undertaken to a limited extent. Potential and currently unused areas of cooperation between the marshal's office and the owners of the wooden buildings could concern, for example, monitoring tourist movements on the WA Route or carrying out surveys of tourists' opinions and preferences. A significant drawback, affecting accessibility, and thus probably the degree of commercialization of the Route's offer, is also the lack of cooperation and arrangements regarding the conditions for making available and visiting the wooden buildings. A significant part of the facilities are not adapted to receive and service tourist movement, both in terms of infrastructure and organization. What's more, detailed descriptions of the monuments in promotional materials attractively designed from the graphic and linguistic viewpoints, do not contain basic information about the hours and conditions of visiting and availability of individual buildings. After all, the first overarching goal of the creation of the WA Route was supposed to be the protection and preservation of the historical wooden buildings, assuming as well that the designated route would become an axis activating economic entities and the local community in activities related to the service of tourist movement, which would appear on the route. As a consequence, the WA Route will become a full integrated tourism product of spatial nature, therefore in addition to attractions (wooden buildings) and routes marked with appropriate signposts, it will also contain accommodations, catering and other infrastructures and amenities expected by tourists. After almost two decades of functioning of the WA Route, it seems that the expected process of developing the route to a full tourism product has not yet occurred. It is not known whether and how the axis designation in the form of route has contributed to the assumed activation and development of other components of the tourism offer in the form of accommodation, catering, etc. This issue may be an interesting area of scientific exploration.

**Author Contributions:** Conceptualization, M.Z.-K. and M.N.; Formal analysis, M.N.; Methodology, M.Z.-K.; Software, M.N.; Supervision, M.Z.-K.; Writing—original draft, M.Z.-K.; Writing—review & editing, M.Z.-K.

**Funding:** This research received no external funding.

**Conflicts of Interest:** The authors declare no conflict of interest.

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
