# Peer review of "The Wooden Architecture Route as an Example of a Regional Tourism Product in Poland"

_sustainability, doi:10.3390/su11185128_

Round 1

Reviewer 1 Report

Dear authors:

Thank you for sharing information about the Wooden Architecture Route. It seems like a very interesting heritage project. The paper was informative but does not meet the criteria to be accepted as a research paper. 

There is no strong purpose of the research. Is it just to present the concept of regional product? Then it is just a descriptive paper which adds very little to the literature.

There isn´t an empirical work as stated in the methods section. For example, what empirical method was used? What "solid analysis" was conducted? What was observed and how these observations were included in the results section?

What does it mean that the empirical work consists of "an analysis of the idea of the creation of the AD route..."?

The literature review on regional product is very repetitive. It basically explains the same points at least three times.

Results: In addition to the wooden structures (attractions), what other characteristics of the region does the route offer, as mention in the literature review? No additional analysis is provided.

It is not clear what the logo of the Route is. 

The paper mentions that the AD Route is extensive in terms of diversity and multiculturalism but no further information is given to support this claim.

Author Response

1. 

The concept of creating regional tourism products consists of a methodological approach integrating these two factors, i.e. tourism and heritage. In recent years, different kinds of touristic routes turned into a popular form of presenting and interpreting the local and regional heritage. They become an essential factor in initiating social and civic change at the local level [1]. The AD route, which is the subject of analysis of the present study, was chosen as an example of regional tourism product since as demonstrated by the research, touristic routes based on a given local heritage are most frequently indicated as examples of regional tourism products [2]. The second important reason for choosing the AD route as an example of regional tourism product is the fact that about 10% of historical monuments in Poland are constituted by wooden buildings and transom structures. Retaining such objects lies in the interest of society for the sake of their historical, artistic and scientific value [1]. The issues associated with the creation and the development of tourism products with the use of the historical, cultural or natural heritage of a given place are important from the point of view of the local and regional development because these products themselves can be thought of as a kind of activation tools of development.

2, 3.

However, for the realization of the latter task, the following guidelines were assumed:

the general, territorial (spatial) way of presenting and defining the tourism product is the closest to what the tourist understands as "product" - therefore the entire experience associated with the arrival and stay in a given place; within the framework of the integrated regional tourism products occurs the activation of the development of the region and the local communities through tourism - in accordance with the principles of sustainable development and the protection of the heritage of the region through its cultivation and preservation for future generations; in the process of creating and shaping a regional tourism product, there are several subjects involved: the private, public, and non-governmental sectors as well as the local community..

On the other hand, in part, the empirical work consists of an analysis of the idea of the creation of the AD route, the structure of the itinerary and its functioning. An analysis of the marketing activities undertaken in connection with the shaping of the tourism product of the AD Route was as well carried out. The primary research method applied is the case study. It’s a qualitative research method, whose main purpose is the most thorough illustration of a certain phenomenon, object, action, or in accordance with the name – specific case. According to A. Lee [3] it is comparable to the natural experiment. Such a method could be applied in many research fields – it is often used, for instance, in law or medicine. Since the 90ies of the last century it is applied in Management sciences as well [4-6] – both as a scientific method and a method of Management education [7] (p.8). Gregorczyk, with reference to Management sciences, defines the case study as an in-depth detailed description, usually real, of a given economic phenomena e.g. the organization, the management process, its elements or business environment, in order to provide conclusions about the causes and results of its course [8] (p. 10). As the author indicates, this method is empirical because it analyzes and evaluates real-world phenomena and is used especially for descriptive research topics. It gives the answer to the question - what, where and how did a given phenomenon happens. This type of scientific research focuses more on an in-depth understanding of the phenomenon than on the analysis of variables. R. Yin recommends using the case study method to find answers to questions that are revealing, and therefore concerning the “how” and “why” a given phenomenon occurs [9] (p. 8).

For the needs of carrying out the case study, various techniques and tools for collecting and analyzing data are used. These can be observations, participatory observations, interviews, surveys, documentation of the organization under study, press sources, Internet sources, databases run by various institutions, etc.. [8] (p. 10).

Reviewer 2 Report

This paper describes the Wooden Architecture Route as an example of a Regional Tourism Product in Poland. While the authors show a good understanding of the subject, the manuscript must be improved:

The objective and relevance of this paper should be clearly stated. The authors provide a description of the Wooden Architecture Route but, Why are this important to academics and/or policy makers? Perhaps contextualizing the study in a wider academic background would improve this point. The methodology used should be discussed and stated. The authors should explain how was the data acquired and analyzed. Literature review: The authors should also reflect on the binomial tourist routes–regional tourism products. The route is organized in different itineraries that articulate numerous tourist resources, as indicated in Table 1. For these itineraries to become authentic regional tourist products, something else is needed. Could the authors provide information on the offer of tourist services, infrastructure and facilities linked to this route? Could the authors provide information on the agents and managers involved on the route?

The following references are recommended:

Gómez-Martín, M.B. (2019). Hiking Tourism in Spain: Origins, Issues and Transformations.  Sustainability11, 3619.

MacLeod, N. (2017). The role of trails in the creation of tourist space. Journal Heritage Tourism, 12, 423–430.

Pearce, D.G. (1992). Alternative tourism: Concepts, classifications, and questions. In Tourism Alternatives: Potentials and Problems in the Development of Tourism; Smith, V.L., Eadington, W.R., Eds.; University of Pennsylvania Press: Philadelphia, PA, USA; pp. 15–30.

Timothy, D.J.; Boyd, S.W. (2015). Tourism and trails: Cultural, ecological and management issues. Channel View Publications: Bristol, UK.

Author Response

The concept of creating regional tourism products consists of a methodological approach integrating these two factors, i.e. tourism and heritage. In recent years, different kinds of touristic routes turned into a popular form of presenting and interpreting the local and regional heritage. They become an essential factor in initiating social and civic change at the local level [1]. The AD route, which is the subject of analysis of the present study, was chosen as an example of regional tourism product since as demonstrated by the research, touristic routes based on a given local heritage are most frequently indicated as examples of regional tourism products [2]. The second important reason for choosing the AD route as an example of regional tourism product is the fact that about 10% of historical monuments in Poland are constituted by wooden buildings and transom structures. Retaining such objects lies in the interest of society for the sake of their historical, artistic and scientific value [1]. The issues associated with the creation and the development of tourism products with the use of the historical, cultural or natural heritage of a given place are important from the point of view of the local and regional development because these products themselves can be thought of as a kind of activation tools of development.

The analysis of the activities and the marketing tools used in terms of the shaping tourism product AD Route indicates the advantage of activities which are promotional and in favour of shaping the image of the product compared to those associated with the creation of a full, integrated offer of the tourism product AD Route or related to its availability. Moreover, in order to respond to the needs and expectations of the tourists, it is necessary to examine their views as well as the opinions and expectations of the local community, whose degree of satisfaction and commitment to create a tourist offer of the place often influences directly the quality of such offer and hence the satisfaction of the tourists. In relation to the management of the AD route, such studies have not been conducted.

An important element testifying the good management of a touristic spatial product in which we acknowledge the AD Route is the cooperation and coordination of activities between all the contributors to the product. In this respect, action is needed to activate and sustain the cooperation of the entities that make up the tourism product discussed. Cooperation with the subjects at the disposal of which are to be found the wooden architecture objects, however difficult - not only because of the multiplicity of such entities, but also as a result of their insufficient or often low involvement - seems to be undertaken to a limited extent. Potential and currently unused areas of cooperation between the marshal's office and the owners of the wooden buildings could concern, for example, monitoring tourist movements on the AD Route or carrying out surveys of tourists' opinions and preferences. A significant drawback, affecting accessibility, and thus probably the degree of commercialization of the Route’s offer, is also the lack of cooperation and arrangements regarding the conditions for making available and visiting the wooden buildings. A significant part of the facilities are not adapted to receive and service tourist movement, both in terms of infrastructure and organization. What's more, detailed descriptions of the monuments in promotional materials attractively designed from the graphic and linguistic viewpoints, do not contain basic information about the hours and conditions of visiting and availability of individual buildings. After all, the first overarching goal of the creation of the AD Route was supposed to be the protection and preservation of the historical wooden buildings, assuming as well that the designated route would become an axis activating economic entities and the local community in activities related to the service of tourist movement, which would appear on the route. As a consequence, the AD Route will become a full integrated tourism product of spatial nature, therefore in addition to attractions (wooden buildings) and routes marked with appropriate signposts, it will also contain accommodations, catering and other infrastructures and amenities expected by tourists. After almost two decades of functioning of the AD Route, it seems that the expected process of developing the route to a full tourism product has not yet occurred.. It is not known whether and how the axis designation in the form of route has contributed to the assumed activation and development of other components of the tourism offer in the form of accommodation, catering, etc. This issue may be an interesting area of scientific exploration.

Round 2

Reviewer 2 Report

Thanks for the reply.